# *Rosa davurica* Inhibited Allergic Mediators by Regulating Calcium and Histamine Signaling Pathways

**DOI:** 10.3390/plants12071572

**Published:** 2023-04-06

**Authors:** Seojun Lim, Sarang Oh, Quynh T. N. Nguyen, Myeongju Kim, Shengdao Zheng, Minzhe Fang, Tae-Hoo Yi

**Affiliations:** 1Graduate School of Biotechnology, Kyung Hee University, Yongin 17104, Republic of Korea; ongane1@khu.ac.kr (S.L.); quynhnguyen@khu.ac.kr (Q.T.N.N.); espritmj@khu.ac.kr (M.K.); mincheol1030@khu.ac.kr (M.F.); 2Snowwhitefactory Co., Ltd., 807 Nonhyeon-ro, Gangnam-gu, Seoul 06032, Republic of Korea; blazma1021@kmu.kr (S.O.); sdjeong0719@khu.ac.kr (S.Z.)

**Keywords:** *Rosa davurica*, allergy, β-hexosaminidase, histamine, Ca^2+^ influx, histidine decarboxylase (HDC)

## Abstract

*Rosa davurica* Pall. exhibits antioxidant, antiviral, and anti-inflammatory properties; however, its pharmacological mechanism in allergy is yet to be understood. This study confirmed the effects of *R. davurica* Pall. leaf extract (RLE) on allergy as a new promising material. To evaluate the therapeutic potential of RLE against allergy, we investigated the effects of RLE on the regulatory β-hexosaminidase, histamine, histidine decarboxylase (HDC), Ca^2+^ influx, nitric oxide (NO), and cytokines induced by lipopolysaccharide (LPS) and DNP-IgE/BSA in Raw 264.7 and RBL-2H3 cells. Furthermore, we examined the effects of RLE on the signaling pathways of mitogen-activated protein kinase (MAPK) and Ca^2+^ pathways. After stimulating Raw 264.7 cells with LPS, RLE reduced the release of inflammatory mediators, such as NO, cyclooxygenase (COX)-2, inducible nitric oxygen synthase (iNOS), interleukin (IL)-1β, -6, and tumor necrosis factor (TNF)-α. Also, RLE reduced the β-hexosaminidase, histamine, HDC, Ca^2+^ influx, Ca^2+^ pathways, and phosphorylation of MAPK in DNP-IgE/BSA-stimulated RBL-2H3 cells. Our studies indicated that RLE is a valuable ingredient for treating allergic diseases by regulating cytokine release from macrophages and mast cell degranulation. Consequently, these results suggested that RLE may serve as a possible alternative promising material for treating allergies.

## 1. Introduction

Allergies develop due to an overreaction of the immune system to extrinsic factors, such as pollen, insect stings, medication, and foods [1]. Acute and chronic exposure to allergic substances contributes to the pathogenesis of allergic rhinitis, atopic dermatitis, allergic asthma, and fatal systemic allergic reaction, termed anaphylaxis [2]. Macrophages are involved in various allergic and autoimmune response mechanisms, secreting Th2-dependent cytokines and reactive oxygen and nitrogen intermediates, exacerbating the severity of allergy and asthma symptoms [3]. As allergen invades tissues, the type-2 helper T cells (Th2) are activated to accelerate the production of immunoglobulin E (IgE), which further cross-links to the high affinity IgE receptor FcεRI on the mast cell surface [4]. Following FcεRI activation, phosphorylation of phospholipase Cγ (PLCγ) is stimulated, promoting the production of diacylglycerol (DAG) and inositol triphosphate (IP3) [5]. Subsequently, this initiates the internal calcium store depletion via the endoplasmic reticulum (ER) resident protein inositol triphosphate (IP3R) and stromal interaction molecule (STIM1) [6]. Extracellular Ca^2+^ ions enter via transient receptor potential canonical (TRPC) and calcium release-activated calcium channel protein 1 (Orai1) channels, which leads to the depletion of Ca^2+^, consequently controlling the fusion of preformed secretory granules and plasma membranes to release histamine [7].

Histamine is one of the earliest identified allergy mediators. The immunoregulation of histamine involves binding with G protein-coupled histamine receptors. Histamine production can be regulated by histidine decarboxylase (HDC), which decarboxylates the amino acid histidine to form histamine [8]. HDC is widely expressed in various body cells, such as gastric mucosa, neurons, parietal cells, mast cells, and basophils [9]. To date, regulating HDC expression is majorly limited to the transcriptional level, which involves several transcription factors, including specificity protein 1 (Sp1), kruppel-like factor 4 (KLF4), GATA binding protein 2 (GATA2), and Microphthalmia-associated transcription factor (MITF). Ai et al. reported that HDC gene expression was downregulated by KLF4 small interfering RNA and KLF4 overexpression [10]. Also, KLF4 inhibited HDC promoter activity by competing with Sp1 in the upstream GC box. Li et al. reported that connective tissue mast cell (CTMC)-specific Gata2 deficient mice failed to develop IgE/mast cell-mediated anaphylaxis [11]. Meanwhile, MITF bound to an enhancer located upstream of the transcription start site of the HDC gene and directed enhancer activity [12]. Thus, targeting HDC expression is a promising alternative to treating allergic diseases. 

*Rosa davurica* Pall. is well known in northeastern Asia as a food source enriched with vitamins C and E [13]. Various therapeutical effects of *R. davurica* have been identified, such as antioxidant, anti-HIV, antiviral, antibiotics, and hypoglycemic activities [14]. Our previous study demonstrated the antioxidative and photoprotective effects of *R. davurica* leaf extract on in vitro human keratinocytes models [15]. The study by Hwang et al. demonstrated the anti-inflammatory and related activity mechanisms of *R. davurica* against Propionibacterium acnes-induced inflammation [16]. However, the effects of *R. davurica* on allergic inflammation and its mode of action on cellular signaling pathways remain unclear. Thus, the current study evaluated the anti-allergic effects of *R. davurica* leaf extract in vitro, using the model of LPS-induced macrophages and DNP-IgE/BSA-sensitized basophils. The results promoted a novel and multitarget strategy for treating allergic inflammation, which utilizes *R. davurica* leaf extract to regulate the mRNA expression of inflammatory mediators of macrophages and inhibit histamine activity by exerting the expression of HDC, MAPK, and calcium influx signaling during the allergic response.

## 2. Results

### 2.1. Chemical Profiling of RLE

The chemical profile method of RLE was used, and the peak patterns were compared at different wavelengths (210, 254, 280, and 360 nm) (Figure 1). As a result, the largest number of components was detected at 280 nm, and ethyl gallate and ellagic acid were detected as major components. The retention times of ethyl gallate and ellagic acid were 16.50 and 19.43 min, respectively, and the retention times of ethyl gallate and ellagic acid detected in the RLE were 16 and 19 min. It was confirmed that the UV spectra of each peak were consistent with the standard materials (Figure 2).

### 2.2. Effects of RLE on Cell Viability

As shown in Figure 3, the induction of either LPS or DNP-IgE/BSA reduced the cell viability by 10.9% and 12.7%, respectively, compared to the untreated group. Besides, RLE did not show significant cytotoxicity in RAW 264.7 and RBL-2H3 cell lines. Thus, RLE at the concentrations of 10, 50, and 100 µg/mL was used as treatment doses for further experiments.

### 2.3. Effects of RLE Extract on NO Production in RAW 264.7 Cells

In normal physiological conditions, NO is a hallmark signaling molecule during inflammatory response; however, its overproduction leads to the pathogenesis of inflammatory diseases, such as asthma, rhinitis, and atopic dermatitis. As shown in Figure 4, NO secretion from macrophage RAW 264.7 significantly increased to 354.6%, compared to the untreated group; however, the treatment of LPS-induced cells with RLE reduced NO production by 40.8% and 43.8% at concentrations of 50 and 100 µg/mL, compared to the LPS control group. The positive control, dexamethasone, exhibited NO production inhibition (reduced by 32.6%) but was ineffective at RLE doses.

### 2.4. Effects of RLE on Cell Degranulation and Histamine Release from RBL-2H3 Cells

As basophils are activated by IgE stimulation, the cells degranulate and secrete β-hexosaminidase, histamine, and several cytokines to recruit other inflammatory cells, including macrophages, eosinophils, and fibroblasts. In this study, the sensitization of RBL-2H3 with DNP-IgE/BSA accelerated the secretion of β-hexosaminidase and histamine by 763.6% and 519.6%, respectively, to the normal cells. However, the supplementation of RLE to the culture medium at doses 50 and 100 µg/mL effectively alleviated β-hexosaminidase and histamine production by 14.7% and 55.6%, and 58.3%, and 73.3%, respectively, compared to the DNP-IgE/BSA group (Figure 5) The effects of RLE at 100 µg/mL was more effective than the positive control tacrolimus, which reduced β-hexosaminidase and histamine by 39.3% and 47.1%, respectively. 

### 2.5. Effects of RLE on Calcium Influx in RBL-2H3 Cells

Ca^2+^ influx positively correlated with histamine production in the basophils. In this study, we observed that stimulation with DNP-IgE/BSA increased Ca^2+^ influx by 1710.2% compared to the normal cells (Figure 6). However, RLE treatment (100 μg/mL) significantly decreased Ca^2+^ influx by 50.0% compared to the DNP-IgE/BSA group.

### 2.6. Effects of RLE Extract on the mRNA Expression of Inflammatory Mediators in RAW 264.7 Cells

In response to inflammation, macrophage cells express various cytokines leading to the vascular permeability and recruitment of inflammatory cells. Under LPS stimulation, the mRNA expression of COX-2, iNOS, IL-1β, IL-6, and TNF- α were increased by 1171.9%, 219.9%, 820.0%, 249.6%, and 290.9%, respectively, compared to the untreated cells (Figure 7) However, RLE treatment reversed these changes by reducing COX-2, iNOS, IL-1β, IL-6, and TNF-α gene expression by 86.3%, 43.5%, 66.5%, 63.5%, and 65.4%, respectively, compared to the LPS control group. 

### 2.7. Effects of RLE on the mRNA and Protein Expression of HDC and HDC Transcription Factors

Histamine was synthesized by the catalyzation of the HDC enzyme, which upregulated its mRNA and protein levels by 110.7% and 52.5% under DNP-IgE/BSA sensitization (Figure 8). However, the treatment of RLE reversed these changes by decreasing the mRNA and protein expression by 61.4% and 54.1%, respectively. 

Although there are still limited findings regarding HDC regulation at the transcriptional level, it was reported that transcription factors SP1 and GATA positively regulate HDC gene expression at the promoter region. Meanwhile, KLF4 negatively controls HDC mRNA levels by suppressing SP1. As shown in Figure 9, the stimulation of DNP-IgE/BSA upregulated the protein expression of SP1 and GATA2 by 117.8% and 98.6%, respectively; meanwhile, it downregulated KLF4 protein levels by 45.0%, compared to the normal cells. However, RLE treatment significantly inhibited SP1 and GATA2 protein levels by 54.6% and 36.1%, respectively, whereas RLE restored KLF4 protein levels by 43.3%, compared to the DNP-IgE/BSA control group.

### 2.8. Effects of RLE on the Protein Expression of MAPK Pathways

The DNP-IgE/BSA sensitization activated MAPK signaling pathways, activating HDC transcription factors, such as SP1 and GATA2. The activation was examined by detecting the phosphorylation levels of MAPKs by western blot analysis. The DNP-IgE/BSA treatment triggered the phosphorylation of p-38, JNK, and ERK by 195.9%, 127.6%, and 183.8%, respectively. However, treatment with tacrolimus and RLE reversed the same, suppressing this phosphorylation. RLE treatment at a concentration of 100 μg/mL significantly inhibited the phosphorylation of p-38, JNK, and ERK by 61.59%, 65.64%, and 60.7%, respectively (Figure 10).

### 2.9. Effects of RLE on the Protein Expression of Calcium Signaling Pathways

The mast cell granulation depends on the concentration of calcium released from the ER. After the release of calcium from the ER, calcium channel proteins, such as STIM1, Orai1, and TRPC1, are activated. Due to this, the influx of extracellular calcium is increased, which accelerates histamine release. Thus, the expression levels of STIM1, Orai1, and TRPC1 were investigated. The stimulation with DNP-IgE/BSA increased the expression of these proteins. In cells treated with RLE, the expressions of STIM1, Orai1, and TRPC1 decreased by 40.6%, 29.4%, and 55.5% at 100 μg/mL (Figure 11). These results indicated that RLE treatment blocked the extracellular calcium influx via the downregulation of calcium channel proteins, including STIM1, Orai1, and TRPC1, thereby reducing histamine release.

## 3. Discussion

It has been previously reported that the pathogenesis of inflammatory diseases, such as atopic dermatitis and asthma, involves direct cell-cell contact between macrophages and mast cells or the secretion of inflammatory mediators. Liu et al. found that human lung macrophages spontaneously secrete “macrophage factors” that interact with mast cell surface-bound IgE to induce calcium-dependent histamine release from human basophils and lung mast cells [17]. On the other hand, mast cell degranulation leads to macrophage recruitment to release inflammatory mediators, such as NO, PGE_2_, and cytokines [18]. Thus, to develop an effective treatment for immune diseases, developing a therapeutic agent that can regulate mast cell and macrophage activation is essential. In this study, an ethanolic extract of RLE demonstrated inhibitory effects on NO production and mRNA levels of COX-2, iNOS, IL-1β, IL-6, and TNF-α in LPS-treated RAW264.7 cells. In addition, RLE reduced degranulation and histamine release from RBL-2H3 cells by downregulating the calcium influx signaling pathway and HDC expression. Therefore, RLE is a promising candidate for an alternative drug and may potentially serve as an anti-allergic agent.

Mast cell degranulation is characterized by histamine release since antigen exposure [19]. Targeting histamine, either preventing its release from the mast cells or use of histaminergic receptor antagonist, becomes part of antihistaminic therapy in allergic diseases [20]. When mast cells are activated by cross-linking membrane-bound IgE with FcεRI, the granules are degranulated, releasing vast amounts of stored mediators, including β-hexosaminidase, histamine, and proteases [21]. Therefore, β-hexosaminidase and histamine are markers for mast cell degranulation. These mediators regulate inflammatory reactions in other immune-related cells, such as macrophages. In DNP-IgE/BSA-stimulated RBL-2H3 cells, the inhibitory effects of RLE on mast cell degranulation were examined. Mast cell degranulation was reduced by suppressing β-hexosaminidase and histamine release (Figure 5). In addition, acute allergic inflammation in LPS-induced macrophages was alleviated following treatment with RLE, resulting in a lower mRNA expression of inflammatory modulators, such as COX-2, iNOS, IL-1β, IL-6, and TNF-α (Figure 7). These findings indicated that RLE acts as an inhibitor for macrophage pro-inflammatory molecules and a “mast cell stabilizer” by inhibiting mast cell-mediator release.

Calcium influx plays a crucial role in inducting degranulation of RBL-2H3 cells, which can regulate the granule-plasma membrane fusion and release of mediators [22]. Also, the reduced calcium concentration in the ER is sensed by STIM1, which promotes the influx of calcium from the extracellular space through calcium channel proteins, such as Orai1 and TRPC1 [23]. Consequently, the cytosolic calcium concentration is elevated, leading to the fusion of preformed granules and plasma membranes. This triggers the release of mediators, such as β-hexosaminidase and histamine, by mast cell degranulation. In this study, RLE significantly reduced the intracellular calcium levels in a dose-dependent manner, compared with only DNP-IgE/BSA-stimulated cells (Figure 6). Additionally, lower expression of calcium channel proteins, such as STIM1, Orai1, and TRPC1, was observed in RLE-treated cells (Figure 11). These results suggested that RLE might suppress the Ca^2+^-dependent degranulation due to the reduction of these calcium channel proteins. 

Various antihistamines are used for treating allergic disorders due to their H1-antagonism [24]. Despite of the availability of abundant antihistamines in the market, developing novel antihistaminic agents with reduced sedation and higher efficiency is essential due to multitargeting. In this study, RLE was demonstrated to act on the mRNA and protein expression of HDC–the sole enzyme that functions in histamine production from histidine (Figure 8). The pharmacological study reported that mRNA and protein expression of HDC was inhibited by mitogen-activated protein kinase (MEK) and c-Jun terminal kinase inhibitor, demonstrating that the MAPK pathway plays a significant role in the transduction of inhibitory signals to suppress HDC expression [25]. The results suggested that the regulatory mechanism of RLE on HDC was mediated by downregulating MAPK signaling, consequently exerting the activity of transcription factors of the HDC gene, including GATA2, MITF, SP1, and KLF4 (Figure 9). Li et al. reported that connective tissue mast cell-specific GATA2 deficient mice were unable to produce IgE/mast cell-mediated anaphylaxis, meanwhile overexpressed MITF significantly restored HDC gene expression in the GATA2 deficient-mast cells [11]. KLF4 inhibited HDC promoter activity via GC-rich sequences in upstream and downstream elements. However, Sp1, a known transcriptional activator with a high affinity for GC boxes, can compete with KLF4 at the HDC promoter to initiate HDC expression [26]. In this study, RLE prevented the phosphorylation of MAPK member proteins, including p38, ERK, and JNK (Figure 10), thus decreasing the expression of GATA2 and SP1 for HDC gene transcription. As a result, KLF4 was upregulated upon treatment with RLE leading to a higher affinity of KLF4 to the HDC promoter to block transcription events.

## 4. Materials and Methods

### 4.1. Reagents

Dulbecco’s modified Eagle’s medium (DMEM), antibiotics (penicillin and streptomycin), fetal bovine serum (FBS), and 0.25% trypsin-ethylenediaminetetraacetic acid (EDTA) were purchased from Gibco BRL (Grand Island, NY, USA). Minimum Essential Medium Eagle, 3-(4,5-dimethylthiazol-2-yl)-2,5-diphenyltetrazolium bromide (MTT), lipopolysaccharide (LPS), 4-nitrophenyl n-acetyl-b-d-glucosaminide (p-NAG), and monoclonal anti-DNP-IgE were supplied by Sigma–Aldrich (St. Louis, MO, USA). DNP-BSA was procured from Invitrogen (Gaithersburg, MD, USA). Primary and secondary antibodies were obtained from Santa Cruz (Santa Cruz, CA, USA), Cell signaling (Danvers, MA, USA), and Bio-Rad Laboratories, Inc. (Hercules, CA, USA). The murine macrophage RAW 264.7 cell line was obtained from KCLB (Seoul, Republic of Korea). The rat basophilic leukemia RBL-2H3 cell line was obtained from ATCC (Manassas, VA, USA).

### 4.2. Sample Preparations

The *R. davurica* leaves were provided by Wonsamrosehip Co., Ltd. (Yongin-si, Gyeonggi-do, Republic of Korea). A total of 100 g *R. davurica* leaves were extracted in 1000 mL of 30% ethanol by shaking on a Twist shaker for 24 h at room temperature. The extract solution was filtered using a 5 μm filter paper (Whatman, Kansas, MO, USA), concentrated in a water bath under vacuum (EYELA WORLD—Tokyo Rikakikai Co., Ltd., Koishikawa, Bunkyo, Japan), frozen, and lyophilized to yield an ethanol extract (RLE). The yield was 20.9%. The voucher specimen was stored in the laboratory at Snowwhitefactory Co., Ltd. (Seoul, Republic of Korea).

### 4.3. Chemical Profiling by High-Performance Liquid Chromatography

High-performance liquid chromatography (HPLC) analyzed the chemical profile of RLE. HPLC analysis was performed in Waters ARC systems using 2998 PDA detectors and Waters Atlantis T3 columns (4.6 mm × 250 mm, 5 μm) (Milford, MA, USA). The flow rate was 1.0 mL/min, and the injection volume was 10 μL. The condition for chromatographic separation was described in Appendix A.

### 4.4. Cell Culture

All cell lines were maintained in an incubator at 37 °C with a humidified atmosphere containing 5% CO_2_. The cells were cultivated in either DMEM media (RAW 264.7 cells) or EMEM (RBL-2H3 cells) containing 10% heat-inactivated FBS and 1% antibiotics and antimycotic medication. The cells were seeded in 96 well plates (RAW 264.7, 1 × 10^6^ cells/well; RBL-2H3, 3 × 10^4^ cells/well) and 60 mm plates (RAW 264.7, 2 × 10^7^ cells/well; RBL-2H3, 6 × 10^5^ cells/plate) until the confluency reached 80%.

### 4.5. Sample Treatment

Both Raw 264.7 and RBL-2H3 cells were treated with different concentrations (1–100 µg/mL) of RLE. For Raw 264.7 cells, they were seeded in a 96-well plate and co-treated with RLE. After 1 h, around 1 µg/mL of LPS was used as a stimulator, and 10 µM of dexamethasone was used as a positive control. For RBL-2H3 cells, they were first sensitized with 50 ng/mL of DNP-IgE for 24 h. Then the cells were pretreated with RLE for 1 h and stimulated with 100 ng/mL of DNP-BSA. Around 50 µM of ketotifen fumarate and 50 ng/mL of tacrolimus were positive controls.

### 4.6. Measurement of Cell Viability

At the end of the incubation period, the MTT reagent was added at a final concentration of 0.1 mg/mL. Next, the MTT-treated cells were incubated for 3 h at 37 °C in a CO_2_ incubator, after which the medium was removed, and 100 μL of DMSO was added to 96 well plates to dissolve the formazan crystals. Finally, the OD was read at 595 nm using a FilterMax F5 microplate reader (Molecular Devices, San Francisco, CA, USA).

### 4.7. Measurement of NO Production

NO production in RAW 264.7 cells was measured using the Griess reagent system (Promega, Fitchburg, WI, USA). A mixture of sulfanilamide solution and N-(1-naphthyl) ethylenediamine dihydrochloride solution (1:1 *v*/*v*) was added to each well and incubated for 10 min at room temperature. The absorbance was recorded at 595 nm using a FilterMax F5 microplate reader (Molecular Devices, San Francisco, CA, USA).

### 4.8. β-Hexosaminidase Release Assay

To investigate the inhibitory effects of RLE on degranulation, the release of β-hexosaminidase was analyzed. After 24 h of seeding, the RBL-2H3 cells were rinsed twice with siraganian buffer, and the cells were incubated with anti-DNP-IgE (50 ng/mL) for 24 h. The IgE-sensitized cells were treated with different concentrations of RLE for 1 h at 37 °C. Then, the cells were stimulated with 100 ng/mL of DNP-BSA and incubated for an additional 4 h under the same conditions. After incubation, the cells were placed in ice for 10 min to terminate the reaction. Around 50 μL of supernatants were collected and mixed with substrate buffer (1 mM p-nitrophenyl-N-acetyl-β-D-glucosamine in 0.1 M citrate buffer, pH 4.5). Following 1 h of incubation, the reaction was stopped with 200 μL/well stop solution (0.1 M Na_2_CO_3_/NaHCO_3_, pH 10.0). The absorbance was read at 405 nm using a FilterMax F5 microplate reader (Molecular Devices, San Francisco, CA, USA).

### 4.9. Histamine Release Assay

Histamine release was quantified with a commercial ELISA kit (Histamine ELISA kit; Abcam, Cambridge, UK). After 2 h of sample treatment and stimulation, cell culture supernatants were collected, and the total concentration of histamine was measured according to the manufacturer’s instructions.

### 4.10. Measurement of Intracellular Calcium Levels

The intracellular calcium levels were analyzed with a commercial Calcium Assay Kit (Colorimetric) (Abcam, Cambridge, UK). After the sample treatment and stimulation process, the cell culture supernatants were collected, and the intra-cellular calcium was measured according to the manufacturer’s instructions.

### 4.11. Reverse Transcription-Polymerase Chain Reaction (RT-PCR)

The total cellular RNA was extracted from the cells using TRIzol (Invitrogen Co., Grand Island, NT, USA). The RNA samples were quantified, and 4 μg of RNA was reverse transcribed using 200 units of reverse-transcriptase and 0.5 μg/μL oligo-(dT)15 dimer (Bioneer Co., Daejeon, Republic of Korea). The amplification was performed using a PCR premix (Bioneer Co., Daejeon, Republic of Korea). The used primers are listed in Table 1. PCR was performed using a Veriti Thermal Cycler (Applied Biosystems, Foster City, CA, USA). PCR products were separated on 2.0% agarose gel electrophoresis and visualized with ethidium bromide.

### 4.12. Western Blot Analysis

The cells and skin tissues were extracted using RIPA buffer. Next, the cells and skin lysates were homogenized to yield equivalent amounts of proteins based on the protein concentration measurements performed with Bradford reagent (Bio-Rad, Hercules, CA, USA). The homogenized proteins were electrophoresed on 10–15% sodium dodecyl sulfate-polyacrylamide gel (SDS-PAGE) and transferred from SDS-PAGE to a nitrocellulose membrane (Amersham Pharmacia Biotech, Buckinghamshire, UK). Non-specific binding was blocked with 5% bovine serum albumin in TBST (50 mmol/1 Tris-HCL, pH 7.5, 150 mmol/1 NaCl, and 0.1% Tween 20) for 1 h at room temperature followed by overnight incubation of membranes with primary antibodies at 4 °C. The membranes were washed with TBST thrice and incubated with a secondary antibody (Santa Cruz Biotechnology Inc., TX, USA) for 1 h at room temperature. Finally, the proteins were visualized using a chemiluminescence detection ECL reagent (GE Healthcare Life science, MO, USA) and quantified with UVI-1D software (UVITEC, Warwickshire, UK).

### 4.13. Statistical Analysis

All data are expressed as the mean ± standard deviation (SD) of three independent experiments. The statistical significance from the control group was evaluated using a one-way analysis of variance, followed by Tukey’s test and *t*-test using Prism software (GraphPad Software Inc., La Jolla, CA, USA).

## 5. Conclusions

In conclusion, this study suggested that RLE could be a valuable ingredient for treating allergic diseases by regulating cytokine release from macrophages and mast cell degranulation. RLE supplements suppression of mRNA expression of inflammatory mediators, including iNOS, COX-2, TNF-α, IL-1β, and IL-6 in LPS-treated RAW264.7 cells. RLE treatment inhibited β-hexosaminidase and histamine release by controlling calcium-related signaling pathways in DNP-IgE/BSA-stimulated RBL-2H3 cells. In particular, a merging strategy for antihistamine was studied, which regulates histidine decarboxylase by exerting MAPK pathways and HDC transcription factors, subsequently decreasing histamine production. Based on the present results, RLE is a natural-derived product. It can be considered a new promising material with fewer side effects than steroids and immunosuppressive drugs, frequently used for treating allergies. However, further studies are essential to verify the anti-allergic effects of RLE in animal models and clinical trials.

## Figures and Tables

**Figure 1 plants-12-01572-f001:**
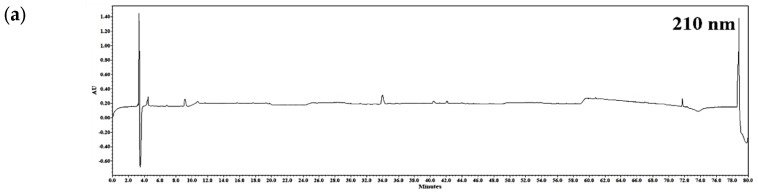
High−performance liquid chromatography (HPLC) profiling of *R. davurica* Pall. leaf extract (RLE) (**a**) 210 nm; (**b**) 254 nm; (**c**) 280 nm; (**d**) 360 nm.

**Figure 2 plants-12-01572-f002:**
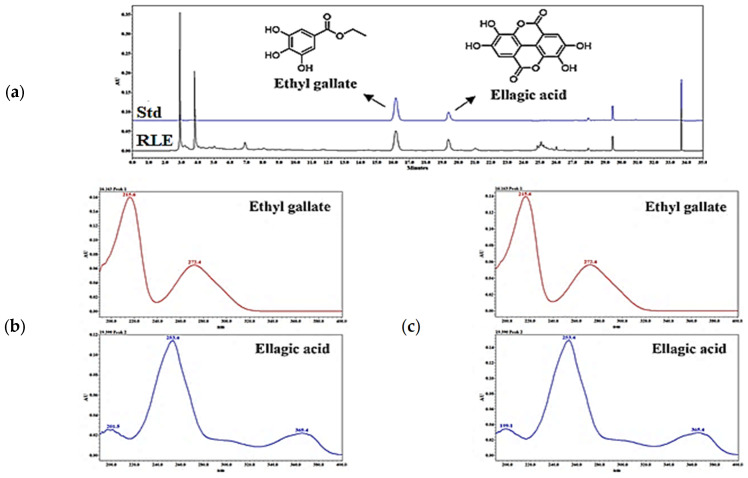
High−performance liquid chromatography (HPLC) chromatogram of the marker compound (**a**) Chromatogram of standard and *R. davurica* Pall. leaf extract (RLE); (**b**) Ultraviolet (UV) spectrum of the standard; (**c**) UV spectrum of RLE.

**Figure 3 plants-12-01572-f003:**
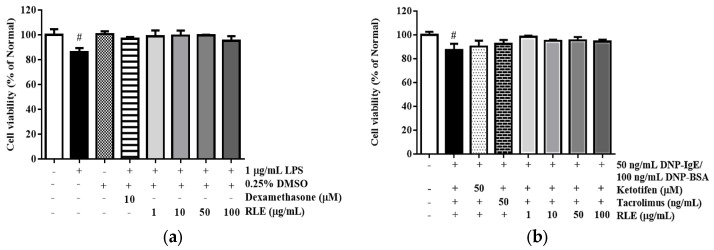
The cytotoxicity of *R. davurica* Pall. leaf extract on RAW 264.7 cells (**a**) and RBL−2H3 cells (**b**). Data are presented as the mean ± SD. # indicates significant differences from the non-treated cells. # *p* < 0.05 vs. the non-treated group.

**Figure 4 plants-12-01572-f004:**
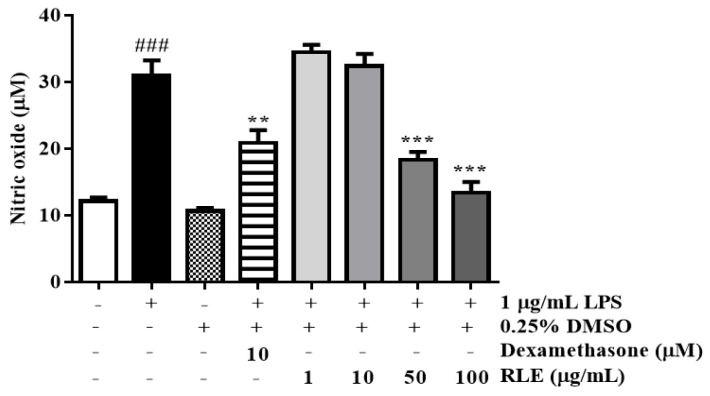
The effects of *R. davurica* Pall. leaf extract on nitric oxide production in LPS−induced RAW 264.7 cells. Data are presented as the mean ± SD. # and * indicate significant differences between the non-treated cells and LPS−induced groups, respectively. ### *p* < 0.001 vs. the non−treated group. ** and *** *p* < 0.01 and 0.001 vs. the LPS−induced control, respectively.

**Figure 5 plants-12-01572-f005:**
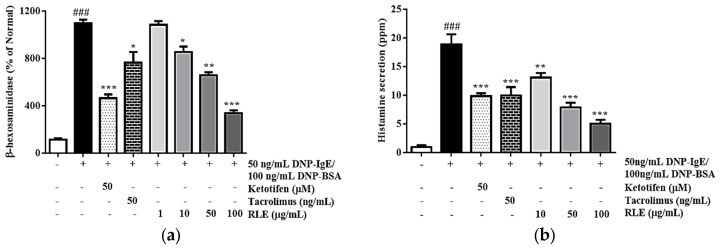
The effects of *R. davurica* Pall. leaf extract on β−hexosaminidase release (**a**) and histamine (**b**) in RBL−2H3 cells. Data are presented as the mean ± SD. # and * indicate significant differences between the non−treated cells and LPS−induced groups, respectively. ### *p* < 0.001 vs. the non−treated group. *, ** and *** *p* < 0.05, 0.01 and 0.001 vs. the DNP−IgE/BSA−sensitized control, respectively.

**Figure 6 plants-12-01572-f006:**
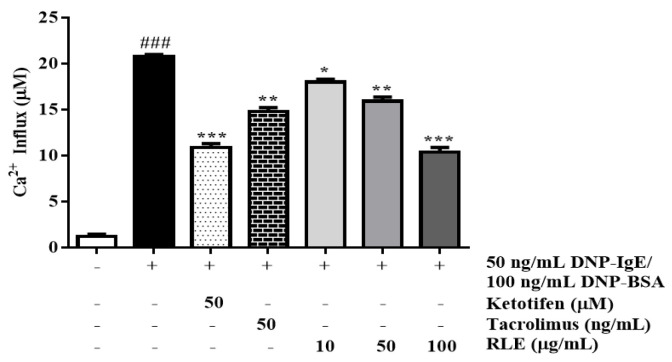
The effects of *R. davurica* Pall. leaf extract on Ca^2+^ influx from RBL-2H3 cells. Data are presented as the mean ± SD. * indicate significant differences between the non-treated cells and DNP-IgE/BSA-sensitized groups, respectively. ### *p* < 0.001 vs. the non-treated group. *, ** and *** *p* < 0.05, 0.01, and 0.001 vs. the DNP-IgE/BSA-sensitized control, respectively.

**Figure 7 plants-12-01572-f007:**
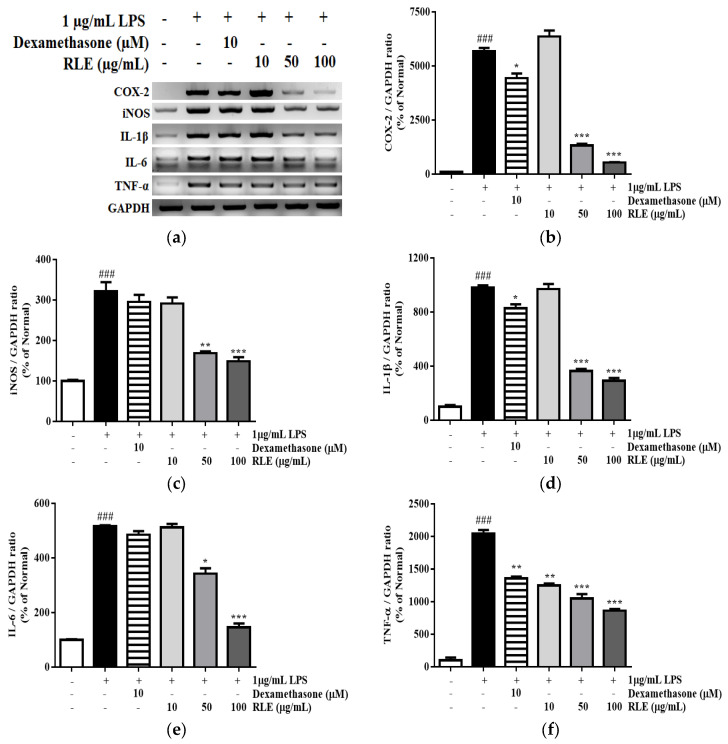
The effects of *R. davurica* Pall. leaf extract on the mRNA levels in LPS-induced RAW 264.7 cells. The mRNA levels were determined by RT-PCR (**a**). The band intensities were quantified by densitometry, normalized to the level of COX−2 (**b**), iNOS (**c**), IL−1β (**d**), IL−6 (**e**), and TNF−α (**f**) calculated as the percentage of the non-treated group. Data are presented as the mean ± SD. * indicate significant differences between the non-treated cells and LPS-induced groups, respectively. ### *p* < 0.001 vs. the non-treated group. *, ** and *** *p* < 0.05, 0.01, and 0.001 vs. the LPS-induced control, respectively.

**Figure 8 plants-12-01572-f008:**
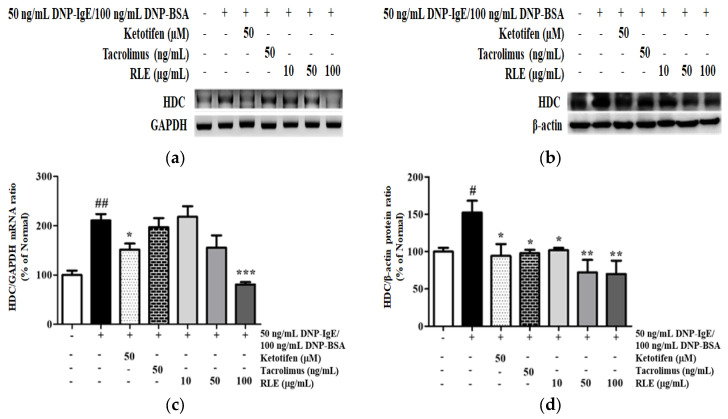
The effects of *R. davurica* Pall. leaf extract on the mRNA and protein levels of histidine decarboxylase (HDC) isolated from RBL−2H3 cells. The mRNA and protein levels were determined by RT-PCR (**a**) and western blot (**b**). The band intensities were quantified by densitometry, normalized to the HDC level of mRNA (**c**) and protein (**d**) calculated as the percentage of the non-treated group. Data are presented as the mean ± SD. # and * indicate significant differences between the non-treated cells and DNP-IgE/BSA-sensitized groups, respectively. # and ## *p* < 0.05 and 0.01 vs. the non-treated group. *, ** and *** *p* < 0.05, 0.01, and 0.001 vs. the DNP-IgE/BSA-sensitized control, respectively.

**Figure 9 plants-12-01572-f009:**
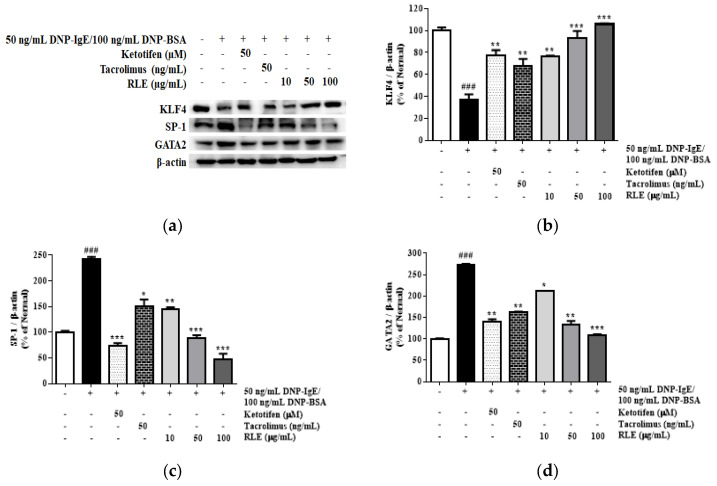
The effects of *R. davurica* Pall. leaf extract on the protein expression of kruppel-like factor (KLF) 4, specificity protein (SP)−1, and GATA2 isolated from RBL−2H3 cells. The protein levels were determined by western blot (**a**). The band intensities were quantified by densitometry, normalized to the level of KLF4 (**b**), SP−1 (**c**), and GATA2 (**d**) calculated as the percentage of the non-treated group. Data are presented as the mean ± SD. * indicate significant differences between the non-treated cells and DNP-IgE/BSA-sensitized groups, respectively. ### *p* < 0.001 vs. the non-treated group. *, ** and *** *p* < 0.05, 0.01, and 0.001 vs. the DNP-IgE/BSA-sensitized control, respectively.

**Figure 10 plants-12-01572-f010:**
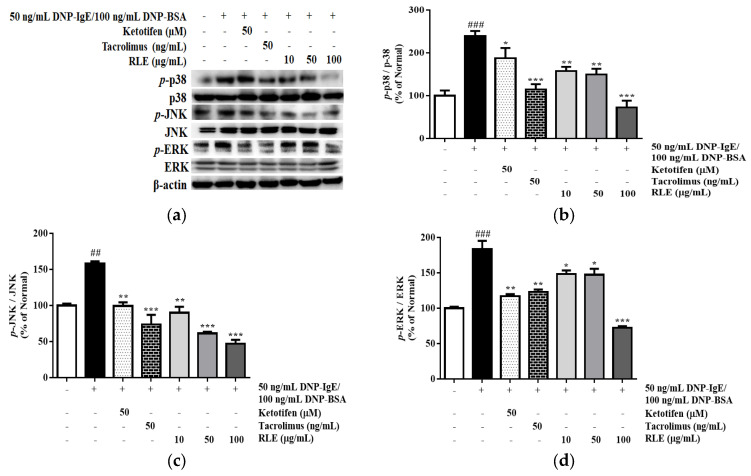
The effects of *R. davurica* Pall. leaf extract on the protein expression of p−38, JNK, and ERK from RBL−2H3 cells. Phosphorylated and non-phosphorylated forms of p−38, JNK, and ERK were detected by western blot (**a**). The band intensities were quantified by densitometry, normalized to the level of p−38 (**b**), JNK (**c**), and ERK (**d**) calculated as the percentage of the non-treated group. Data are presented as the mean ± SD. # and * indicate significant differences between the non-treated cells and DNP-IgE/BSA-sensitized groups, respectively. ##, ### *p* < 0.01, and 0.001 vs. the non-treated group. *, ** and *** *p* < 0.05, 0.01, and 0.001 vs. the DNP-IgE/BSA-sensitized control, respectively.

**Figure 11 plants-12-01572-f011:**
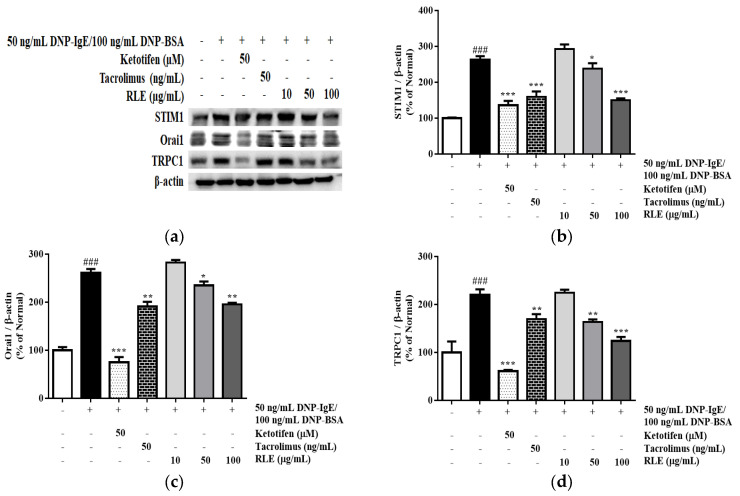
The effects of *R. davurica* Pall. leaf extract on the protein expression of STIM1, Orail1, and TRPC1. The protein levels were determined by western blot (**a**). The band intensities were quantified by densitometry, normalized to the level of STIM1 (**b**), Orail1 (**c**), and TRPC1 (**d**) calculated as the percentage of the non-treated group. Data are presented as the mean ± SD. * indicate significant differences between the non-treated cells and DNP-IgE/BSA-sensitized groups, respectively. ### *p* < 0.001 vs. the non-treated group. *, ** and *** *p* < 0.05, 0.01, and 0.001 vs. the DNP-IgE/BSA-sensitized control, respectively.

**Table 1 plants-12-01572-t001:** List of primers for RT-PCR.

Primer ^1^		Sequence (5′-3′)
Rat GAPDH	Sense	TGA TGA CAT CAA GAA GGT GGT GAA G
	Anti-sense	TCC TTG GAG GCC ATG TAG GCC AT
Rat COX-2	Sense	ACT CAC TCA GTT TGT TGA GTC ATT C
	Anti-sense	TTT GAT TAG TAC TGT AGG GTT AAT G
Rat iNOS	Sense	CCT CCT CCA CCC TAC CAA GT
	Anti-sense	CAC CCA AAG TGC TTC AGT CA
Rat IL-1β	Sense	TGC AGA GTT CCC CAA CTG GTA CAT C
	Anti-sense	GTG CTG CCT AAT GTC CCC TTG AAT C
Rat IL-6	Sense	CTG CAA GAG ACT TCC ATC CAG
	Anti-sense	AGT GGT ATA GAC AGG TCT GTT GG
Rat TNF-α	Sense	TCT CAT CAG TTC TAT GGC CC
	Anti-sense	GGG AGT AGA CAA GGT ACA AC
Mouse GAPDH	Sense	TGA TGA CAT CAA GAA GGT GGT GAA G
	Anti-sense	TCC TTG GAG GCC ATG TAG GCC AT
Mouse HDC	Sense	TGA CAA CTT CTC ACT CCG AGG
	Anti-sense	ACA AGG TTA GCA GCC TCT CG

^1^ Primer design based on the NCBI/Primer-BLAST tool with standard parameters.

## Data Availability

The data presented in this study are available in this paper.

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
