# Peer review of "Rosa davurica* Inhibited Allergic Mediators by Regulating Calcium and Histamine Signaling Pathways"

_plants, 2023, doi:10.3390/plants12071572_

Round 1

Reviewer 1 Report

The topic of the present work is interesting: the role of many plant extracts in allergic diseases still need pharmacological validation. Here the authors can find suggestions to improve the organization of the paper and the quality of result’s presentation.

Title: Rosa davurica inhibited “Inhibited allergic mediators” by regulating calcium and histamine signaling pathways 

Abstract: since this work is an in vitro investigation, I suggest less emphasis in the “therapeutic potential”. The bioavailability of the extract and the role in specific allergic diseases (by in vivo investigation) is still required.

Introduction: line 63 – several words regarding the organ from the plant used by traditional medicine may help to understand the rational of using leaf (are leaf the commonly used organ?). Moreover, the previous literature regarding the chemical composition and available extracts from Rosa davurica should be provided here. 

Methods

Line 294. The analytical standards used for the identification should be reported here. How were the specific wavelengths selected? The method exclude the detection of compounds with higher absorption (for example, anthocyanins)..

Line 309: 10^6 cells in 96 well/plate is quite a lot; did the authors plated for immediate confluency?

Line 313: the sensitization of RAW cells is not clear; were RAW cells treated with PMA to induce the differentiation and adhesion?

Line 326: was NO measured in cell media?

Line 317: the protocol of sensitization is different from line 335. It is not clear to which outcome measure the sensitization reported in line 317 is referred. 

Line 345: the protocol for histamine induction is not clear

Line 351: IgE anti-DNP? I suggest to uniform the description of sensitization in the paragraph related to cell stimulation and avoid repetitions (if the protocols are the same, it should be).

Line 374: primary antibodies and their targets should be precisely described. There is a wide use of western blot in this work, thus antibodies need to be well described.

Results

In general, results can be better organized and described. The cell line and the time of treatment should be added in each caption. It can be also useful to describe the results obtained in RAW cells and then in RBL cells, since there are no common parameters. Otherwise, the cell line can be confounded during the lecture. 

-       The chemical analysis is poorly described. Figure 2 is impossible to read. The chromatogram of RLE reported in Fig.2a is different from Fig.1 (280 nm), but the authors stated that analytical standards were detected at 280nm, is it correct? 

Moreover, the presence of flavonoids is reported by the literature: did the authors performed any tentative identification of other compounds? At least, the quantification of total phenols and flavonoids, if polyphenols are the major components, can be performed by simple colorimetric assays.

-       Figue 3: the inclusion of DMSO 0.25% in the experiments is not clear. Is it the vehicle? In this case, it should be added to the stimulatory condition also.

-       Line 102: the role of NO in the pathogenesis of diseases is more limited. It is involved in the pathogenesis. It should be better to show the effect on iNOS expression immediately after this data.

-       Line 127: the comparison with tacrolimus is uncorrect. It acts at nM level, while extracts act at uM level. Tacrolimus is clearly superior. To this purpose, Tacrolimus should be expressed as nM (like Ketotifen).

-       Figure 7. The description of Fig. 9a (WB summary) is missing in the caption. Are the graph referred to the mRNA or WB analysis?

-       Figure 8. The western blot analysis of HDC/GADPH was reported on the graph relative to mRNA. Were both analysis conducted (PRC and WB on HDC/GADPH?) in addition to HDC/b-actin by WB? The font of Figure 8 should be uniformed.

-       Figure 9. The description of Fig. 9a (WB summary) is missing in the caption.

Discussion

A final graph or a “graphical summary” may help the comprehension of the mechanism involved and the cell line context.

Line 220. The work does not demonstrate that RLE reach the targets in complex biological systems. The authors should comment the limits of the present work (in vitro, bioactive compounds partially known…).

Line 235. The effect on RAW was already cited in line 218.

The effect of ellagic acid and gallic acid derivatives in allergy was previously reported. The author omitted this aspect. For example, a recent review summarized the biological studies in allergic diseases (in vitro, in vivo models) (PMID: 36364420).

Reviewer 2 Report

The work has significant content to be published and is very well reasoned, easy to read and sufficiently illustrated and should be nominated for publication.

Line 291-292

100 g of fresh leaves yielded 20.9 g of extract?

Or were the leaves dehydrated?

How was the moisture of the leaves?
